# Mitochondrial ATP synthesis is essential for efficient gametogenesis in *Plasmodium falciparum*
Penny C. Sparkes[1], Mufuliat Toyin Famodimu[1], Eduardo Alves[1], Eric Springer [2], Jude Przyborski [2] & Michael J. Delves [1] ✉

*Plasmodium* male and female gametocytes are the gatekeepers of human-to-mosquito transmission, therefore essential for propagation of malaria within a population. Whilst dormant in humans, their divergent roles during transmission become apparent soon after mosquito feeding with a rapid transformation into gametes – males forming eight motile sperm-like cells aiming to fertilise a single female gamete. Little is known about how the parasite fuels this abrupt change, and the potential role played by their large and elaborate cristate mitochondrion. Using a sex-specific antibody and functional mitochondrial labelling, we show that the male gametocyte mitochondrion is less active than that of female gametocytes and more sensitive to antimalarials targeting mitochondrial energy metabolism. Rather than a vestigial organelle discarded during male gametogenesis, we demonstrate that mitochondrial ATP synthesis is essential for its completion. Additionally, using a genetically encoded ratiometric ATP sensor, we show that gametocytes can maintain cytoplasmic ATP homeostasis in the absence of mitochondrial respiration, indicating the essentiality of the gametocyte mitochondrion for transmission alone. Together, this reveals how gametocytes responsively balance the conflicting demands of a dormant and active lifestyle, highlighting the mitochondria as a rich source of transmission-blocking targets for future drug development.

Malaria is a life-threatening parasitic disease to which nearly half the world's population is at risk. Despite global efforts towards eradication, malaria remains a significant public health problem. In 2022, it caused an estimated 249 million cases and 608,000 deaths[1].

*Plasmodium*, the causative agent of malaria is transmitted to and from vertebrate hosts through the bite of *Anopheline* mosquitoes. To adapt to these two very different environments, the parasite undergoes a complex life cycle[2]. Whilst the *Plasmodium* asexual cycle is responsible for disease pathology, it is only the gametocyte stage which is infectious to mosquitoes. With every asexual cycle, a proportion of parasites undergo an alternative sexual developmental pathway to form male and female gametocytes. These gametocytes sense uptake by the mosquito through a temperature decrease and the mosquito-derived activating factor xanthurenic acid[3], then rapidly differentiate into male and female gametes within 15–20 min. Unlike non-*Laverania Plasmodium* species, gametocytes of *P. falciparum* which causes the deadliest form of malaria, have an extended developmental period of 8–10 days. This is divided into five morphological stages (Stages I–V), with only (mature) Stage V gametocytes infectious to mosquitoes. Gametocytes have no control over when they will be ingested by a mosquito during blood feeding. Therefore, they are adapted to maximise their lifespan in the human host whilst being always ready to rapidly respond to stimuli inducing gametogenesis upon entering the mosquito. Mature *P. falciparum* gametocytes are estimated to have a mean circulation time in peripheral blood of 4.6–6.5 days[4], however as a population, gametocytes can be infectious to mosquitoes for significantly longer[5].

Mature gametocytes are insensitive to most antimalarials[6] which enables them to escape drug treatment targeting the pathogenic asexual stages and leaves them free to transmit and propagate within a population. The cellular mechanisms behind this insensitivity are not well understood, but likely are caused by the divergent biology of gametocytes compared to asexual blood stage parasites and the relative dormancy of the gametocyte stage[6,7]. One intriguing difference between asexual blood stage parasites and mosquito stage parasites is how they regulate energy metabolism. Asexual parasites exclusively generate adenosine triphosphate (ATP) through glycolysis and show a minimal mitochondrial activity which is linked to the regeneration of cofactors essential for pyrimidine biosynthesis rather than

[1]Department of Infection Biology, London School of Hygiene and Tropical Medicine, London, UK. [2]Biochemistry and Molecular Biology, Interdisciplinary Research Center, Justus Liebig University, Giessen, Germany. ✉e-mail: michael.delves@lshtm.ac.uk

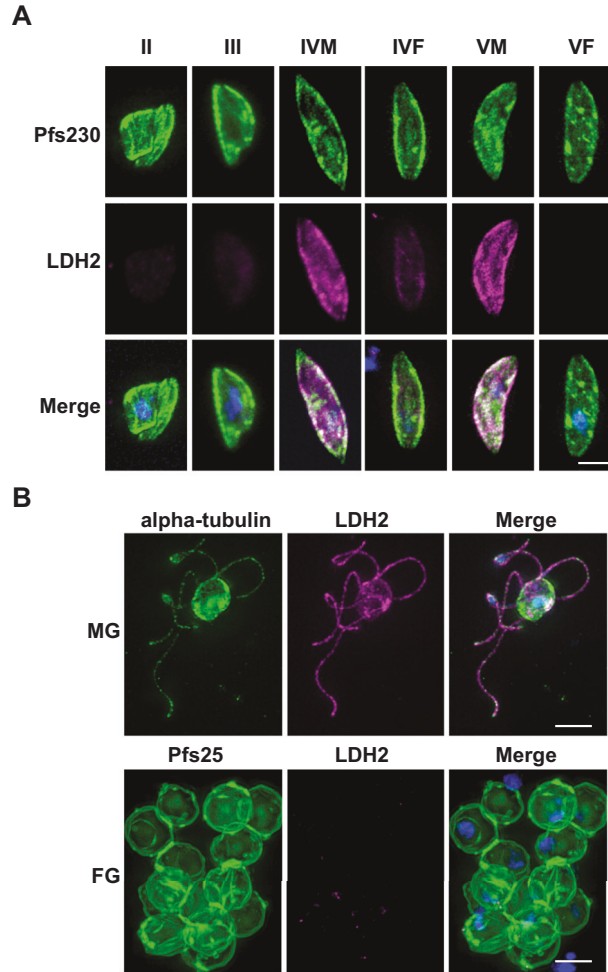

**Fig. 1 | Anti-LDH2 is a specific marker for P. falciparum male gametocytes and gametes. A** Immunostaining with anti-LDH2 (magenta) showed specific staining of a subpopulation of Pfs230-positive (green) stage IV and stage V gametocytes. **B** Male gametes stained with anti-alpha tubulin II (green) also were LDH2-positive, whilst Pfs25-positive (green) female gametes were LDH2-negative. Merged images also show DAPI (blue). Scale bars = 5 µM.

ATP production through oxidative phosphorylation[8]. Indeed, in *P. falciparum*, six out of eight genes of the tricarboxylic acid (TCA) cycle are dispensable for asexual development[9]. To date, two genes from the TCA cycle have been deleted from parasite lines normally competent for mosquito transmission. Deletion of aconitase produced parasites that failed to form mature gametocytes and deletion of α-ketoglutarate dehydrogenase produced parasites with phenotypically normal gametocytes that failed to form oocysts when fed to mosquitoes[9]. Further evidence for the essentiality of mitochondrial respiration for parasite survival in the mosquito is given by the potent transmission-blocking activity of antimalarials targeting mitochondrial energy metabolism such as atovaquone[10,11]. In preparation for transmission, gametocytes develop a large cristate mitochondrial network[12]. Reflecting this, proteomic, transcriptomic and metabolomic data shows an upregulation of TCA and mitochondrial capacity, particularly in female gametocytes[13]. In a dormant cell, increased capacity for ATP production appears unnecessary and potentially detrimental to longevity by increased generation of reactive oxygen species. Conversely, given the essentiality of mitochondrial respiration for onward mosquito transmission, mitochondrial expansion in the gametocyte appears to be essential. However, antimalarials targeting the mitochondria show little or no activity upon gametocytes[6], reinforcing the hypothesis that mitochondrial expansion is in preparation for transmission rather than important for their survival in the human host.

During mosquito transmission, male and female gametes fertilise in the gut of the mosquito to form a zygote. The female gamete contributes the bulk of cellular biomass to the zygote including the mitochondrion[14]. Male gametes consist of little more than genomic material and a microtubule-rich motile flagellum[15], powered by glycolysis alone[16], which seek out the female. As the parasite mitochondrion is female-inherited, it is currently unclear why the male gametocyte would also invest resources in developing an extensive mitochondrial network.

Through sex-specific antibody labelling, chemical inhibition of mitochondrial oxidative phosphorylation and quantitative imaging, we demonstrate that male gametocytes require mitochondrially-derived ATP to adequately fuel the rapid and energetic transformation that occurs during male gametogenesis. Our results support the essential role of the parasite mitochondrion during transmission and indicate that the estimated ~445 mitochondrial proteins of *Plasmodium*[17] may be a rich source of transmission-blocking drug targets.

## Results

### PfLDH2 (*PF3D7_1325200*), a predicted lactate dehydrogenase is specifically expressed in Stage IV/V male gametocytes and male gametes

Previous reports using sex specific markers or transgenic reporter parasites have noted ambiguity of marker expression, especially in Stage IV gametocytes that have not yet reached final maturity. For example, Schwank and colleagues found that antibodies against alpha tubulin II, whilst upregulated in mature male gametocytes, also showed significant immunoreactivity in Stage IV female gametocytes[18]. Furthermore, Lasonder and colleagues observed that immature transgenic gametocytes expressing GFP under the dynein heavy chain promoter (male) and mCherry under the PfP47 promoter (female) expressed both transgenes[13]. To minimise this ambiguity, we searched the literature to identify a marker that could identify male gametocytes with greater certainty.

*PF3D7_1325200* and its *P. berghei* orthologue *PBANKA_1340400* (Rel 4) have been shown to be specifically and highly expressed in male gametocytes at the protein[13,19] and transcript level[20]. *PF3D7_1325200* (hereafter referred to as LDH2) is annotated as a putative lactate dehydrogenase, although its specific function or role is currently unknown. To complement other existing mouse monoclonal and rabbit polyclonal markers in gametocytes, we raised a dual peptide rat antibody against LDH2. Anti-LDH2 showed intense surface-localised staining in immunofluorescence assays (IFAs) against a sub-population of Stage IV and V gametocytes (Fig. 1A). No immunoreactivity was observed in earlier gametocyte stages. Recent single cell transcriptomic data indicates that gametocyte sexual dimorphism is apparent only after Stage III of gametocyte development[21] which supports the male gametocyte-controlled expression of LDH2. Co-staining with an antibody reactive for the female gametocyte-specific protein PfG377, showed absolute inverse correlation with LDH2-positive gametocytes (Supplementary Figs. 1 and 2) further supporting LDH2 expression as a marker for male gametocytes. Furthermore, intense punctate anti-LDH2 staining in exflagellating male gametes was observed which interestingly did not appear to co-localise with alpha tubulin II within the cell (Fig. 1B). Finally, a complete absence of anti-LDH2 staining from Pfs25-positive female gametes was also observed (Fig. 1B). Consequently, this establishes that LDH2 expression is a specific marker for Stage IV and V male gametocytes and microgametes, and can be used to accurately distinguish between male and female gametocytes.

### Quantitative imaging of male and female gametocytes

Having established the specificity of anti-LDH2, the differential immunoreactivity was utilised to determine the sex of mature stage V gametocytes and compare their morphological differences by quantitative immunofluorescence. Gametocytes were prelabelled with the dye MitoTracker™ Red CMXRos (hereafter, MitoTracker) which accumulates in functional mitochondria possessing a negative membrane potential and is retained after aldehyde fixation. After fixation, cells were co-stained with anti-LDH2

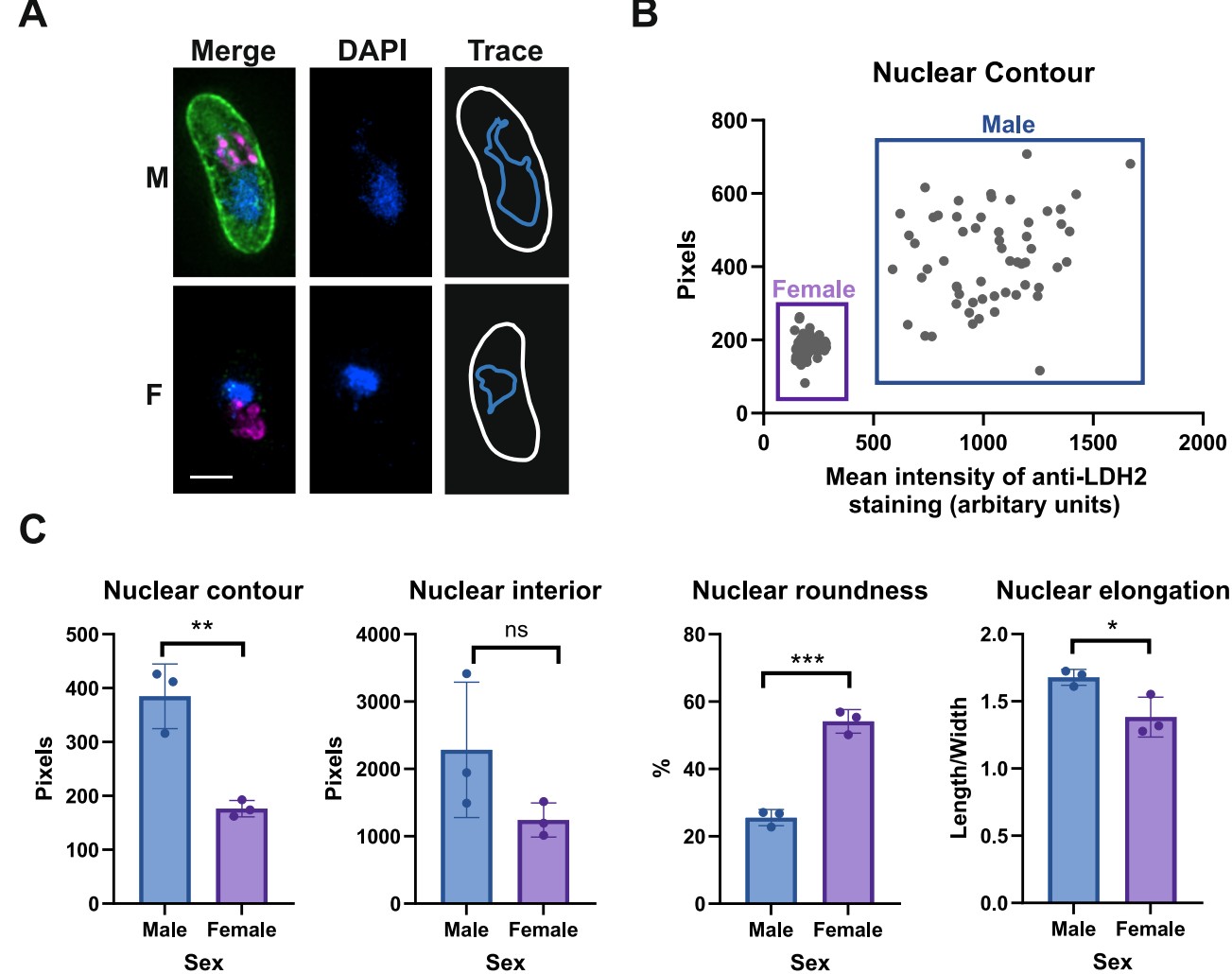

**Fig. 2 | Contrasting the differences between nuclear organisation of male and female gametocytes using quantitative imaging. A** Anti-LDH2 (green) was used to discriminate between male (LDH2-positive) and female (LDH2-negative) gametocytes colabelled with MitoTracker™ CMXRos (magenta) and DAPI (blue). Cell regions of interest (ROI) were manually traced, and nuclear ROI subsequently automatically traced using thresholding of DAPI staining intensity. Scale bar = 3 µM. **B** Using the cell ROI, mean anti-LDH2 staining per gametocyte was calculated and plotted against the calculated nuclear contour of the corresponding nucleus. Two distinct populations of cells with low LDH2 immunoreactivity (females) and high LDH2 immunoreactivity (males) were observed ($n = 139$ gametocytes). **C** A variety of nuclear parameters were calculated and compared between populations of male and female gametocytes. $n = 3$ independent experiments of 133, 68, and 139 gametocytes measured. Unpaired Student's $T$-test: ns = $p > 0.05$; *$p > 0.05$; **$p < 0.01$; ***$p < 0.001$. Degrees of freedom (df) = 4. Error bars denote standard error of the mean (SEM).

and DAPI so that gametocyte sex, and nuclear and mitochondrial organisation could be compared. Gametocytes were imaged at random, identified, and then manually assigned individual "regions of interest" (ROI) that traced their outline. An automated thresholding algorithm was then used to automatically draw ROIs around the nucleus and mitochondria in each cell (Figs. 2A, 3A and Supplementary Fig. 3). Morphological parameters for each gametocyte nucleus and mitochondria were then automatically calculated and extracted. Parameters measured included contour (perimeter of the ROI), interior (area of the ROI), roundness (100% being perfectly round), and elongation (1 being not elongated; >1 indicating a cell longer in one axis over another). Gametocyte sex was assigned by calculating the mean fluorescence intensity of anti-LDH2 over the entire gametocyte ROI (Fig. 2A). When visualised on a scatterplot, two distinct populations of cells emerged – one tightly clustered population with very low levels of anti-LDH2 staining, and one more diffuse cluster with high levels of anti-LDH2 staining (Fig. 2B). The 'low' population were assumed to be female gametocytes and the 'high' population, males.

Using our LDH2-based assignment of sex, the 'male' gametocyte population had a significantly larger nuclear contour than the 'female' population ($p = 0.0043$, unpaired Student's $T$-test) and was more elongated

($p = 0.0327$, unpaired Student's $T$-test) (Fig. 2C). Conversely, the nuclei of the 'female' gametocyte population were significantly rounder than those of the 'male' population ($p = 0.0003$, unpaired Student's $T$-test). It is known that female gametocytes possess a relatively compact, rounded nucleus, whilst male gametocytes adopt a larger, more relaxed, elongated nuclear morphology hypothesised to be important for access to the genome by the DNA replication machinery during gametogenesis[22]. This demonstrates that our data is in concordance with previous studies, further supports LDH2 as a marker that can confidently discriminate between male and female gametocytes and validates the quantitative imaging approach for comparing bulk morphological differences in gametocytes.

We then reanalysed the gametocyte image dataset again, focusing on mitochondrial morphology (Fig. 3A). Previously it has been observed that gametocyte mitochondrial morphology is highly diverse and is not linked to sex or maturity[23]. However, a recent study using high resolution focused ion beam milling - scanning electron microscopy (FIB-SEM) has reported that the male gametocyte mitochondrial network is generally elongated and stretches a significant distance through the gametocyte, whilst the female mitochondrial network is generally observed in a localised cluster[22]. Similarly, we frequently observed males with an elongated network and females

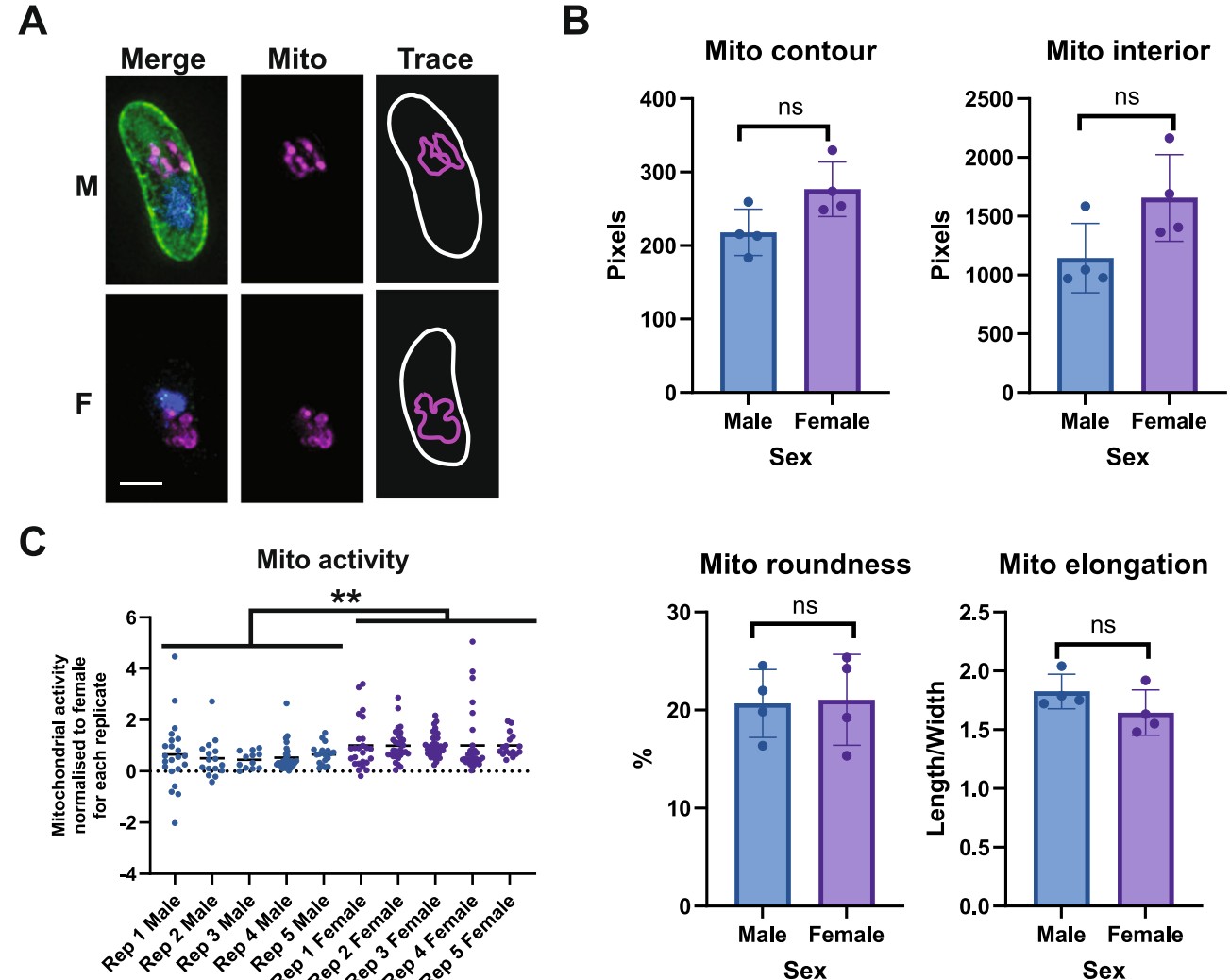

**Fig. 3 | Contrasting the differences between mitochondrial organisation of male and female gametocytes using quantitative imaging. A** Anti-LDH2 (green) was used to discriminate between male (LDH2-positive) and female (LDH2-negative) gametocytes co-labelled with MitoTracker$^{TM}$ CMXRos (magenta) and DAPI (blue). Scale bar = 3 μM. **B** The same dataset used to study gametocyte nuclear morphology was reanalysed similarly using MitoTracker staining to generate mitochondrial ROIs for each cell. A variety of mitochondrial parameters were calculated and compared between populations of male and female gametocytes. $n = 4$ independent experiments of 133, 111, 48, and 147 gametocytes measured. Unpaired Student's $T$-test: ns

$= p > 0.05$. df = 6. Error bars denote standard deviation. **C** Labelling for 25 min with a non-saturating concentration of MitoTracker (12.5 nM) was used to compare mitochondrial activity of male and female gametocytes ($n = 5$ independent experiments of 47, 47 49, 59, and 35 gametocytes measured). Data presented here is normalised to the mitochondrial activity of the female gametocytes for each replicate, black bar indicates the mean of the replicate. A statistical comparison (Ratio paired t-test) was performed before normalisation and showed that the difference in activity between gametocyte sex over all replicates was significant ($p = 0.002$. df = 4). On average, male gametocytes accumulated 44.4% less MitoTracker than females.

with a clustered mitochondrial network (Supplementary Fig. 4). However, this did not translate into statistically significant differences in male and female mitochondrial contour, interior, roundness or elongation when gametocytes were measured randomly and in large numbers (Fig. 3B). We hypothesise that this either may be due to the limitations of observing a complex three-dimensional structure in two dimensions, or alternatively may simply reflect the heterogeneity/dynamic structure of male and female gametocyte mitochondrial networks.

We then focused on the mitochondrial activity of male and female gametocytes by quantifying the total MitoTracker staining per gametocyte. The manufacturer recommended MitoTracker concentration of 200 nM gave highly oversaturated accumulation over a 25 min incubation which, while suitable for studying morphology, was unsuitable for quantification of activity. 12.5 nM MitoTracker was determined to be sufficient for labelling gametocyte mitochondria without oversaturation and was used in all subsequent experiments (Supplementary Fig. 5). Absolute levels of mitochondrial staining intensity varied between independent experimental replicates,

however male gametocytes on average possessed 44.4% less mitochondrial activity than females (Ratio-paired $t$-test; $p = 0.002$) (Fig. 3C).

### The effect of mitochondrial inhibitors on male and female gametocytes

Whilst we did not observe significant differences in male and female gametocyte mitochondrial morphology using our experimental methodology, the observed reduction in accumulation of MitoTracker in the male mitochondrion is consistent with a reduced mitochondrial membrane potential and reduced mitochondrial activity. We then investigated whether male and female gametocyte mitochondria show different or similar susceptibilities to mitochondrial inhibitors. Mature Stage V gametocytes were incubated for 24 h with atovaquone (inhibiting the $Q_0$ site of cytochrome BC1), ELQ-300 (inhibiting the $Q_i$ site of cytochrome BC1), DSM-265 (inhibiting *P. falciparum* dihydroorotate dehydrogenase (DHODH)), and oligomycin A (inhibiting the $F_0$ subunit of ATP synthase). Atovaquone and ELQ-300 both showed a dose-dependent inhibition of male and female mitochondrial

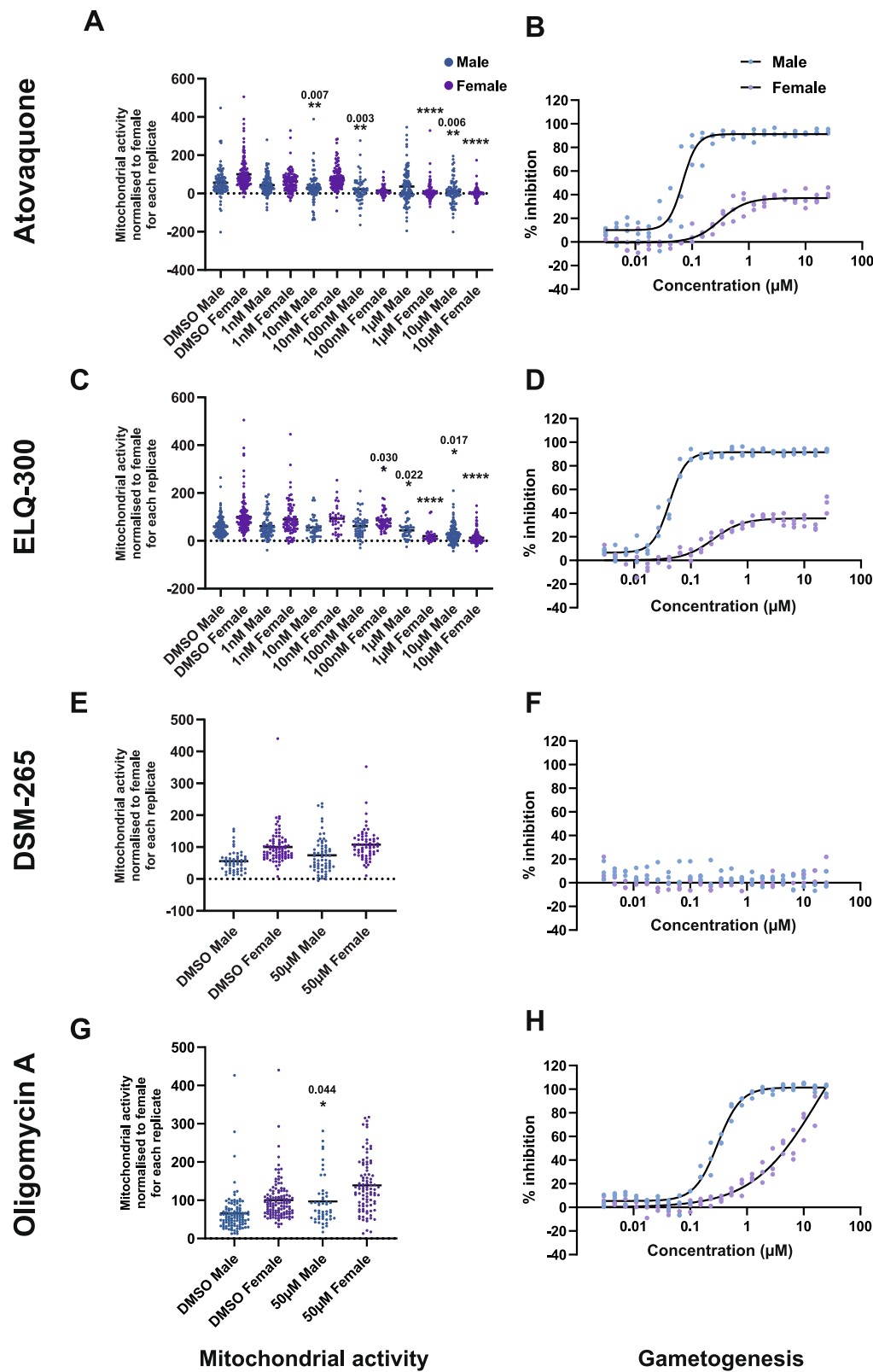

activity that became statistically significant as drug concentration increased (unpaired Student's *t*-test) (Fig. 4A + C). Despite the mitochondrial activity of male gametocytes being initially lower than females, there was no consistent pattern suggesting the mitochondrion of either sex was affected more under drug treatment than the other sex. However, given that the male gametocyte mitochondria appear to have lower baseline activity than females,

it may be possible that even with a similar inhibitory effect, males reach a critical activity threshold before females.

In addition to studying mitochondrial activity, the functionality (i.e., their ability continue development and to form gametes) of male and female gametocytes treated similarly for 24 h with the same mitochondrial inhibitors was tested in the *P. falciparum* Dual Gamete Formation Assay

**Fig. 4 | The effect of mitochondrial inhibitors on male and female gametocyte mitochondrial activity and ability to form gametes.** The mitochondrial activity of male and female gametocytes treated with inhibitors for 24 h (**A, C, E, G**) were calculated following a 25 min incubation with 12.5 nM MitoTracker, fixing and staining with anti-LDH2. Male gametocytes were identified by LDH2 immunoreactivity. Mitchondrial activity was calculated by quantyifying the background-corrected amount of Mito-Tracker fluorescence within the gametocyte ($n = 70–294$ gametocytes per replicate, 3–5 independent experiments). Graphed data presented here is individual cells for all replicates that have been within-replicate normalised to the mean value of female DMSO. Black bar denotes the mean value. Statistical tests show unpaired Student's $T$-test of normalised values. Tests performed against male gametocyte data were normalised to the female gametocyte DMSO control of each replicate and vice versa for the tests for female data. Stars and number above data indicate statistical significance and $p$ value compared to the DMSO control (**** = $p ≤ 0.0001$). Absence of a star indicates not significant change. df = 8, 8, 4, 8 respectively for atovaquone, ELQ-300, DSM-265 and oligomycin A. The functional viability of male and female gametocytes was quantified in the Pf DGFA (**B, D, F, H**) ($n = 3$ independent experiments. Datapoints of each replicate are plotted with black line showing the calculated 4 parameter dose response curve used to determine the $IC_{50}$ value). Mitochondrial inhibitors used: (**A, B**) Atovaquone; (**C, D**) ELQ-300; (**E, F**) DSM-265; (**G, H**) Oligomycin A.

(Pf DGFA)[24]. Both atovaquone and ELQ-300 gave dose-dependent inhibition of male gametogenesis with $IC_{50}$s of 68.5 nM and 42.2 nM respectively (Fig. 4B + D). Interestingly, inhibition of female gametogenesis gave classical dose response curves, however both did not plateau at 100% inhibition, but plateaued at 37.2% and 35.6% for atovaquone and ELQ-300 respectively. The female gamete readout in the Pf DGFA is surface expression of the female gamete marker Pfs25. We hypothesise that this observed partial plateau in the female Pf DGFA readout could potentially be due to atovaquone and ELQ reducing mitochondrial activity in female gametocytes/gametes which could cause a general decrease in metabolic activity and consequently reduced protein synthesis. In the absence of mitochondrial activity, glycolysis alone potentially could fuel some expression of Pfs25, causing the early inhibition plateau falling short of 100% inhibition. This intermediate Pfs25 expression could manifest in a proportion of female gametes falling below the limit of assay detection in the Pf DGFA which is what is observed. Whether or not this compromises the fertility of female gametes is unclear.

In accordance with published reports that DSM-265 is not transmission-blocking and lacks activity in the Pf DGFA[25], we observed no inhibition against male and female gametocytes in the Pf DGFA even at 25 µM (Fig. 4F). Reflecting this, 50 µM DSM-265 did not affect the mitochondrial activity of male and female gametocytes (Fig. 4E). DHODH, the target of DSM-265 is one of five dehydrogenases able to feed reduced ubiquinone into the electron transport chain[26] and so it is likely that this redundancy accounts for lack of inhibitory activity. Oligomycin A at 50 µM increased the mitochondrial activity in both male and female gametocytes (Fig. 4G), however only the increase in male mitochondrial activity was statistically significant. We hypothesise that rather than oligomycin A enhancing mitochondrial activity per se, what we observe is greater accumulation of MitoTracker within the gametocyte mitochondrion caused by inhibition of the ATP synthase proton channel causing hyperpolarisation of the mitochondrial membrane potential. However, whilst not depolarising the gametocyte mitochondrial network, we observe dose-dependent inhibition of male gametogenesis by oligomycin A with an $IC_{50}$ of 303.9 nM (Fig. 4H) suggesting that inhibiting mitochondrial ATP generation impacts the viability of male gametocytes. We observed increasing inhibition of female gametocytes, however, this did not reach a maximal plateau by 25 µM, suggesting that female gametogenesis is less affected.

## Inhibition of mitochondrial respiration by oligomycin A stalls male gametogenesis

Having observed that chemical disruption of mitochondrial activity impacted the ability of gametocytes to form gametes (particularly males) after 24 h preincubation, we investigated whether this effect was due to a specific requirement of the mitochondrion during gametogenesis or a side-effect of prolonged mitochondrial inhibition resulting in declining gametocyte viability. Male gametogenesis is a rapid (and presumably energy-demanding) process. It involves shape change (from a falciform, crescent shape to round = "rounding up"), three endomitotic genome replications, assembly of eight microtubule-rich flagellae, and egress from the erythrocyte[27]. It has previously been established that glucose uptake during male gametogenesis is essential for successful exflagellation[28] suggesting that the male gametocyte does not rely on an internal store of energy to fuel gametogenesis but rather generates ATP on demand. However, the relative contribution of ATP from glycolysis and oxidative phosphorylation to this process is unknown. To better understand this question, we first needed to optimise gametogenesis conditions. Glucose is present both in the RPMI-based culture/exflagellation media, and the added human serum component. Typically, gametogenesis is induced in xanthurenic acid-containing medium containing 10–20% serum, and it has been previously reported that successful exflagellation requires serum concentrations greater than 1.5%[29]. To minimise confounding glucose contributed by the human serum component, we performed a titration of human serum in exflagellation medium both in the presence and absence of glucose in the RPMI medium (Fig. 5A). Crucially, we observed that 10% serum in glucose-free RPMI was sufficient to drive exflagellation at a level indistinguishable from the control (Fig. 5A, Supplementary Fig. 6). As serum concentration declined, glucose-free RPMI was less able to support exflagellation, and exflagellation was absent in glucose-free RPMI plus 0.1% serum. Conversely, in glucose containing RPMI, exflagellation was maximal at all serum concentrations tested, even 0.1%. In the complete absence of serum, exflagellation in glucose containing RPMI was still observed, however, it became highly inconsistent between independent replicates suggesting that there is an additional serum factor(s) required for efficient exflagellation. Taken together, we selected 0.1% serum as the optimal condition for onward experiments which permitted maximal exflagellation in the presence of glucose but does not support any exflagellation in the absence of exogenous glucose.

Taking this further, we then titrated down the glucose concentration of our culture medium (containing 0.1% serum) and determined that the glucose concentration that gives 50% of maximal exflagellation was 19 mg/L (105 µM) and maximal exflagellation achieved by ~52 mg/L (290 µM) (Fig. 5B). Given that the Mitotracker labelling and Pf DGFA assays (Fig. 4) were performed with 4 g/L glucose (77-fold in excess), we conclude that glucose availability is not a limiting factor in our assays and therefore glycolysis alone is not able to compensate for inhibition of mitochondrial respiration.

Using 2 g/L glucose gametocyte medium with 0.1% human serum, we investigated whether mitochondrial ATP synthesis is required during gametogenesis. Gametocytes were washed in warm (Albumax-free) gametocyte culture medium containing 0.1% human serum (i) with 2 g/L glucose; (ii) without glucose; (iii) with 2 g/L glucose plus 5 µM oligomycin A; then gametogenesis was triggered in condition-matched exflagellation medium. As expected, 20 min later, male gametocytes showed high levels of exflagellation in the presence of glucose and no exflagellation in the absence of glucose (Fig. 6A). If glucose was present during gametogenesis but mitochondrial ATP generation inhibited by 5 µM oligomycin A, exflagellation was significantly reduced by 89.5% (Ratio-paired $t$-test; $p = 0.007$). This indicates that an active mitochondrial ATP synthase is needed during male gametogenesis for efficient exflagellation.

In parallel to studying the overall viability of male gametocytes in the presence and absence of glucose and oligomycin A, samples of each treatment condition in Fig. 6A were fixed 20 min after triggering gametogenesis (when exflagellation of the glucose-containing negative control sample could be expected to be maximal) and visualised by fluorescence microscopy (Fig. 6B). Intriguingly, upon stimulating gametogenesis in the absence of glucose, male gametocytes still 'round up' – transforming from their resting crescent shape to spherical. Thereafter, they arrested development and did not form flagellated gametes as observed in the glucose negative control.

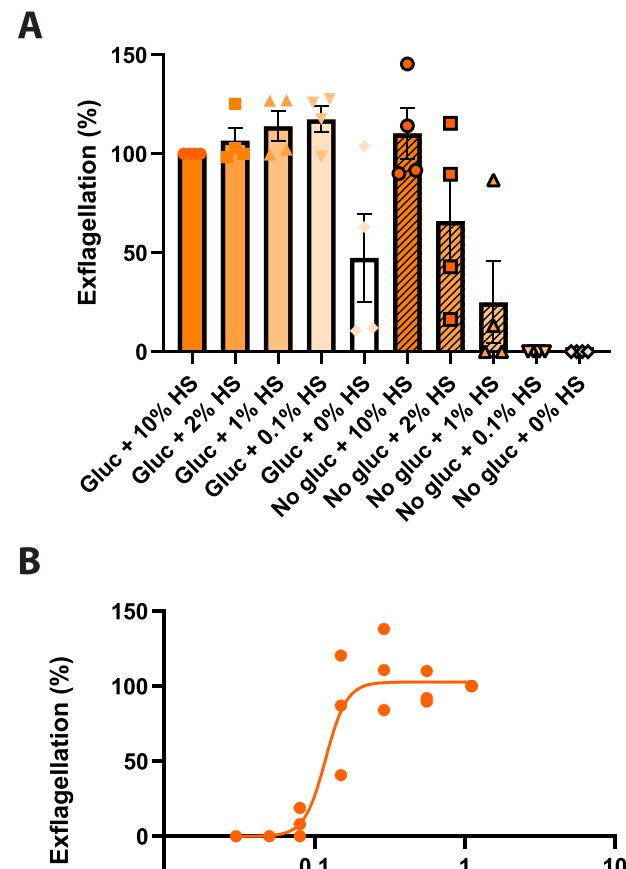

**Fig. 5 | Investigating the dependency of exflagellation on human serum and glucose. A** Mature gametocytes were washed in culture medium containing varying concentrations of human serum, with or without the standard RPMI glucose concentration of 2 g/L. Then, gametogenesis was triggered in condition-matched exflagellation medium. Exflagellation was then quantified 20 min later by automated microscopy. Data is normalised to the control conditions of glucose plus 10% human serum. $n = 4$ independent experiments. Error bars denote standard error of the mean. **B** Mature gametocytes were washed in culture medium containing a fixed concentration of 0.1% human serum and varying concentrations of glucose (0.1–0 g/L). Then, gametogenesis was triggered in condition-matched exflagellation medium. Exflagellation was then quantified 20 min later by automated microscopy. Data is normalised to the control conditions of glucose plus 10% human serum. $n = 3$ independent experiments. Datapoints of each replicate are plotted with the orange line showing the calculated 4 parameter dose response curve used to determine the $IC_{50}$ value.

This early halt to male gametogenesis seems show an apparent lack/reduction of DNA replication (only faint DAPI staining) and disordered microtubule arrangement. As glucose is only withdrawn from the gametocytes immediately prior to triggering gametogenesis, we hypothesise that this phenotype represents the extent of male gametogenesis possible utilising only the immediate pool of ATP contained within the cell if no exogenous source of glucose is available to replace it. Male gametocytes activated in the presence of glucose and oligomycin A appear to proceed further in development than those in the absence of glucose, but also arrest before exflagellation is complete (Fig. 6B, Glucose + 5 µM OA). Under these conditions, most adopted a spherical shape with more defined microtubule structures, but showed small, focal spots of DAPI staining consistent with the absence/reduction of DNA replication. Occasionally, cells were observed with a single large 'super-flagellum'[30] but still lacked evidence of DNA replication or flagellated male gametes with normal morphology. Whether these aberrant gamete structures are capable of limited motility and so responsible for the

residual level of "exflagellation" detected in the automated microscopy assay (Fig. 6A) is unclear, however it is unlikely that these are fertile male gametes capable of supporting onward development in the mosquito.

## Glycolysis alone can sustain ATP homeostasis in gametocytes under glucose-replete conditions but mitochondrial respiration buffers ATP homeostasis under glucose-scarce conditions

To determine the effect of mitochondrial disruption by oligomycin A on cytoplasmic ATP abundance in gametocytes, we generated a parasite line episomally expressing a cytoplasmically-localised Förster resonance energy transfer (FRET) sensor ATeam1.03YEMK[31,32]. The sensor consists of cyan fluorescent protein (CFP) coupled to yellow fluorescent protein (YFP) by a peptide linker derived from a bacterial $F_1F_0$ ATP synthase ε subunit. This allows ratiometric measurement of changing ATP abundance within a cell by comparing the FRET fluorescence emission intensity of ATP-bound vs ATP-unbound sensor. ATP sensor expressing gametocytes were washed three times in different gametocyte culture medium conditions (glucose-containing or glucose-free medium both without human serum or Albumax, and with or without 5 µM oligomycin A) warmed to 37 °C to prevent induction of gametogenesis. Cells were then washed again with warm condition-matched PBS with/without glucose and oligomycin A, allowed to settle on warmed slides and imaged by fluorescence microscopy in a pre-warmed environment chamber. The fluorescence of individual gametocytes (maintained in a non-activated state at 37 °C) was then observed microscopically at 1 min intervals starting at 3 min post transferral to the respective medium formulations (Fig. 7).

Gametocytes maintained in PBS + glucose displayed a FRET ratio of ~18 arbitrary units - which only slightly declined over the 20 min observation period. A similar gradual decline has been previously observed in ATeam1.03YEMK-expressing trophozoites by Springer et al. and attributed to gradual photobleaching of the sensor[32]. To establish a baseline at which intracellular ATP abundance reaches a minimum value or falls below the sensitivity of the sensor, gametocytes were maintained in glucose-free PBS for 120 min. Imaging of these glucose-deprived gametocytes yielded a mean FRET ratio of 9.77 ( ± 0.01 SEM). Gametocytes maintained in glucose-free PBS and monitored over time showed a progressive decline in FRET ratio indicative of gradual depletion of cytoplasmic ATP. In the 23 min observation window, these gametocytes only reached approximately two thirds of the way to the established 120 min baseline indicating that ATP depletion in gametocytes is gradual. In contrast, it is reported that trophozoites deplete ATP to the sensor baseline in approximately 10 min if glucose utilisation is inhibited by excess 2-deoxyglucose[32]. These observations support the notion that mature Stage V gametocytes are metabolically less active than asexual stages. To investigate whether mitochondrial ATP synthesis is necessary for gametocytes to maintain ATP homeostasis, the FRET ratio of gametocytes maintained in PBS-glucose plus 5 µM oligomycin A was studied (Fig. 7). Oligomycin A treatment made no observable difference to the gametocyte FRET ratio when compared to the glucose-only control. This implies that under glucose-replete conditions, glycolysis alone is sufficient maintain cytoplasmic ATP homeostasis in mature Stage V gametocytes. However, in the absence of glucose, additional inhibition of mitochondrial ATP synthesis by oligomycin A suppressed the gametocyte FRET ratio substantially faster than withdrawing glucose alone, reaching the baseline FRET ratio in only 10 min.

## Discussion

Males and female gametocytes play different roles during transmission. Whilst superficially morphologically similar, it is well established that male and female gametocytes express substantially different repertoires of proteins and show different sensitivities to antimalarial molecules[13,33]. Underlying this is the divergent cell biology that male and female gametocytes employ to fulfil their individual roles to enable parasite sexual reproduction and onward infection of the mosquito. The ability to distinguish between male and female gametocytes confidently and unambiguously is essential to study their differential cell biology. Although its cellular function is yet to be

**Fig. 6 | The quantitative and morphological differences in gametogenesis in the presence of mitochondrial inhibitor oligomycin A.**
**A** Gametogenesis was triggered in ookinete medium containing 0.1% human serum, with (2 g/L "Glucose") or without glucose ("No glucose"), or glucose and 5 μM oligomycin A ("Glucose + OA"). Exflagellation was counted 20 min later using automated microscopy to capture exflagellation centres and automated image analysis for quantification. Data presented here is the mean of five independent experiments normalised to the exflagellation activity of the "Glucose" sample for each replicate. However statistical comparison (Ratio paired $t$-test) was performed before normalisation and showed that the difference in activity between "Glucose" and "Glucose + OA" samples is statistically significant ($p = 0.007$. $n = 5$. df = 4). Error bars denote standard error of the mean. **B** Parallel samples of the cells were fixed 20 min post induction of gametogenesis and stained with anti-alpha tubulin and DAPI to visualise microtubule and DNA morphology. Bar = 5 μm.

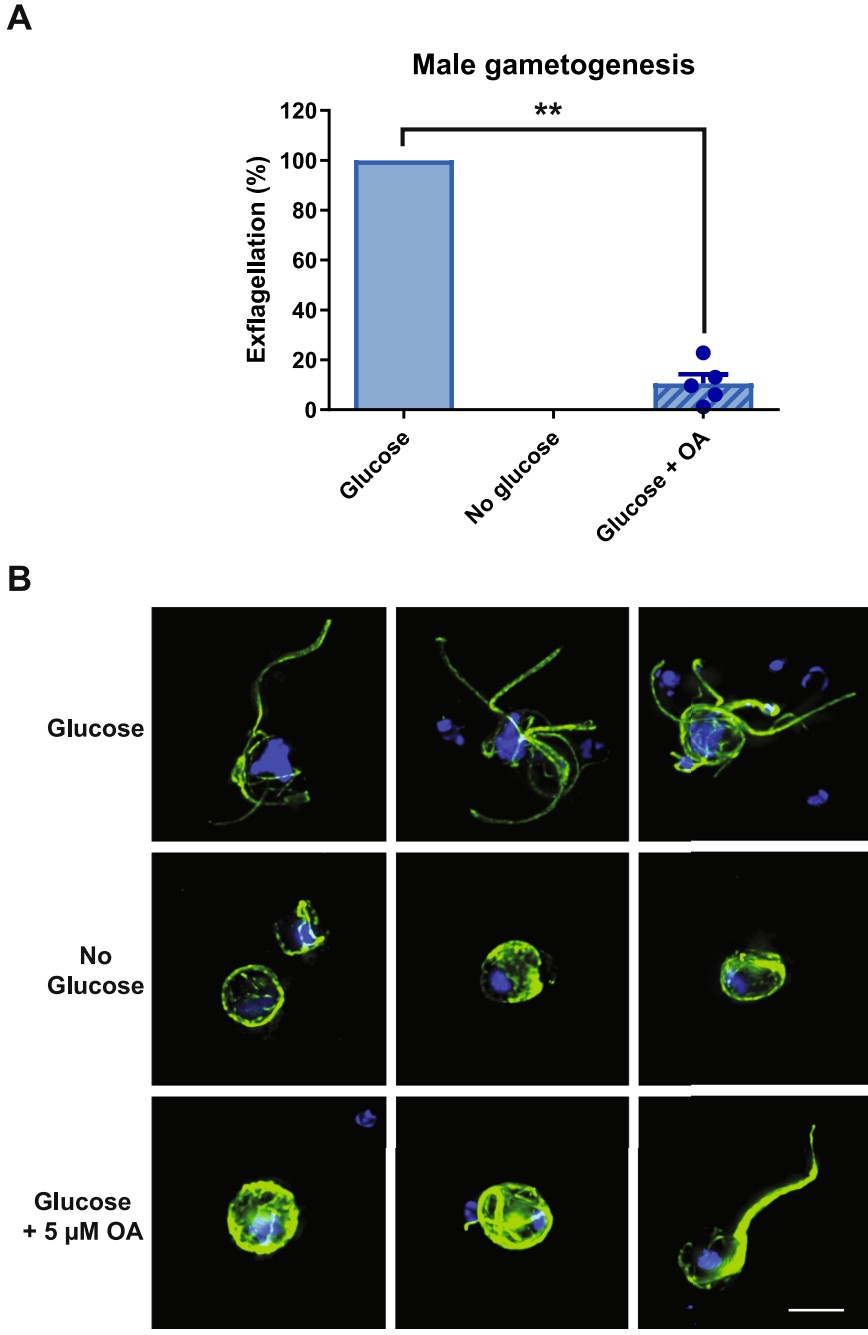

elucidated, LDH2 expression provides a potent tool in our toolbox for the identification of male gametocytes.

In this study, we have shown that LDH2-positive male gametocytes accumulate less mitochondrial-localised MitoTracker dye than female gametocytes. Given that we did not observe significant differences in the overall size and morphology of the male and female mitochondrion, we conclude that our findings suggest that the male mitochondrion is less active than the female. This is supported by previous studies which have shown that proteins involved in mitochondrial respiration are more abundant in female gametocytes[13]. The obvious reason for mitochondrial investment in female gametocytes is to prepare the parasite for onward obligate mitochondrial respiration in the mosquito host. However, this does not explain why male gametocytes, which lose their mitochondrion during gametogenesis, still invest resources transforming their small acristate asexual mitochondrion into an elaborate cristate structure. Through inactivating the mitochondrial respiratory chain with chemical inhibitors, we

have demonstrated a crucial role for mitochondrial function during male gametogenesis. Our data suggests that whilst dispensable in quiescent mature stage V gametocytes, a functional mitochondrion is essential to progress gametogenesis through to completed exflagellation (Fig. 8). Male gametogenesis is clearly a highly energetic process, involving complex signalling pathways, DNA replication and complete remodelling of the cell to form eight gametes. We observed that simultaneously triggering gametogenesis and inhibiting mitochondrial ATP synthesis rapidly arrests this process soon after the gametocyte rounds up but before DNA replication has started (or is significantly underway). We speculate that as well as a general lack of cytoplasmic ATP throughout gametogenesis, ATP depletion could cause an accumulation of errors early in the carefully choreographed sequence of events of male gametogenesis which could also prevent the formation of viable gametes.

Intriguingly, male gametes do not possess a mitochondrion[16]. Currently, it is not clear at which point a male gametocyte with a mitochondrial

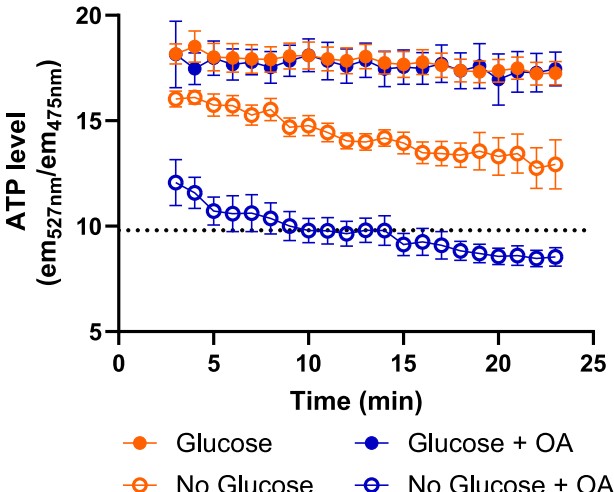

**Fig. 7 | Ratiometric measurement of cytoplasmic ATP using a FRET-based sensor ATeam1.03YEMK.** Gametocytes were washed in culture medium three times with or without glucose and with or without 5 μM oligomycin A. Finally, gametocytes were then pelleted in phosphate buffered saline (PBS) containing matched glucose/oligomycin A conditions and imaged over time at 37 °C. Imaging commenced at 3 min post treatment to account for processing time and time taken for cells to settle onto the slide for imaging. Dotted line indicates the FRET value recorded for gametocytes incubated in PBS alone for 120 min. Closed orange circles indicate gametocytes incubated with 2 g/L glucose; open orange circles indicate gametocytes incubated without glucose. Closed and open blue circles indicate gametocytes incubated with and without glucose respectively in the presence of 5 μM oligomycin A. Data represents the mean of 6–9 independent replicates performed pairwise with all possible combinations of treatment. Error bars denote the standard error of the mean (SEM).

network becomes a mass of mitochondria-less male gametes. Future work is needed to identify the fate of the male mitochondria. Potentially it could remain active throughout male gametogenesis to fuel the process and then remain discarded in the residual body as gametes emerge. Alternatively, an active mitochondrion may only be essential for several events during gametogenesis and then either be surplus to the needs of the parasite or recycled by mitophagy (which to our knowledge has not yet been observed in *Plasmodium*) and its raw materials used to form the mature gametes. The effect of temperature on gametocyte mitochondrial activity is also an unstudied factor that may play a role in male gametogenesis. Gametogenesis is, in part, stimulated by a temperature decrease. Temperature is well known to affect the rate of biological processes. Potentially, as gametocytes cool from 37 °C in the human host to ambient temperature in the mosquito gut, the rate of parasite glycolysis decreases to such an extent that it cannot keep up with demand for ATP. In mammalian systems, a decrease in mitochondrial respiration rates caused by hypothermia is counteracted by an increase in efficiency of respiration thought to be linked to reduction of proton leak across the mitochondrial membrane[34,35]. It is tempting to speculate that such a mitochondrial system could also play a role in optimising ATP production in the parasite under the cooler conditions of the mosquito.

Our observation that intracellular ATP homeostasis in gametocytes does not appear to be affected by mitochondrial inhibition indicates that antimalarial molecules targeting the mitochondria are unlikely to demonstrate a phenotype in gametocytes. This may lead to them being overlooked in current transmission-blocking antimalarial drug screens. However, as displayed by the molecules tested in this study, they likely are potently transmission blocking even before the infection is established in the mosquito. *Plasmodium* possesses an estimated 445 mitochondrial proteins[17]. Even if only a fraction of these proteins are essential, this still represents a substantial pool of targets for antimalarial drug discovery. Targeting gametogenesis rather than gametocyte viability will require molecules that possess a long serum half-life to be present during initial

uptake of gametocytes by the mosquito or cause irreversible damage to the mitochondrion.

## Materials and methods
### Generation of polyclonal antibodies against LDH2 and PfG377
Polyclonal antibodies against the male gametocyte marker LDH2 (PF3D7_1325200) and the female gametocyte marker PfG377 (PF3D7_1250100) were commercially raised in rats and guinea pigs respectively by Eurogentec using the 2 peptide 28-Day Speedy programme. For anti-LDH2, rats were immunised with a peptide corresponding to the C terminus of the protein (C + NYKHETVVDDENKPN-cooh) and an internal peptide (C + HDFRKDLPKGRALDI). Peptides all contained an added cysteine residue at the N terminal. Immunogenic peptides were selected at the manufacturer's recommendation. Similarly, for anti-PfG377, guinea pigs were immunised with peptides internal to the sequence (C + PQSAQNYDRNKFSGT and C + LSDDHDIDSKEYTEN).

### Asexual blood stage parasite culture
*P. falciparum* NF54 strain asexual blood stage parasites were cultured in *P. falciparum* custom medium (RPMI-1640, 2.3 g/L sodium bicarbonate, 2 g/L glucose, 5.9 g/L HEPES, 50 mg/L hypoxanthine, 1% Albumax II, 5% human AB+ serum, 30 mg/ml L-glutamine) and type O, A or AB human blood at 37 °C under 3% $O_2$/5% $CO_2$/92% $N_2$ (malaria gas) Culture medium was replenished daily. Cultures were passaged to 0.3% parasitaemia and 4% haematocrit every 2–3 days. Culture parasitaemia of asexual maintenance cultures was kept under 5% to prevent premature gametocyte commitment and maintain high capacity to generate gametocytes when induced. All parasites were PCR-tested monthly to confirm absence of Mycoplasma.

### Gametocyte culture
The gametocyte culture protocol was adapted from Delves et al.[24] and the below recipe for gametocyte medium was used for all experiments unless deviations are explicitly stated. Gametocyte cultures were seeded from asexual blood stage cultures at 2% parasitaemia and 4% haematocrit in gametocyte medium (RPMI-1640, 2.78 g/L sodium bicarbonate, 4 g/L glucose, 5.9 g/L HEPES, with 5% human AB+ serum, 0.5% Albumax, 3.7% 100xHT supplement (ThermoFisher), 30 mg/ml L-glutamine,) at 37 °C under malaria gas. 3/4 of spent culture medium was replaced daily from gametocyte cultures with fresh gametocyte medium for 14 days. All surfaces, reagents and culture media were warmed to 37 °C to prevent premature activation of gametocytes. At day 14 of culture, maturity of gametocytes was confirmed by triggering gametogenesis in a sample and quantifying exflagellation (male gametogenesis). Gametogenesis was triggered by cooling a sample of culture to room temperature, adding xanthurenic acid (100 μM final concentration) and transferring to a haemocytometer. Cultures showing at least 0.2% exflagellation (of total cells) were used for subsequent experiments (mature day 14 gametocytes hereafter).

### Generation of an NF54 line episomally expressing ATeam1.03YEMK
The plasmid expressing ATeam1.03YEMK was a kind gift from Dr Jude Przborski[32]. Although the plasmid was designed for stable integration using *attB-attP* recombination, this site was lacking from our unmodified NF54 strain. Therefore, the plasmid was transfected by the erythrocyte pre-loading electroporation method[36] and maintained as an episome under constant blasticidin S selection. We observed that the plasmid is not tolerated well by the parasite and their asexual replication rate was low. To generate the high parasitaemia needed to induce gametocyte commitment, it was necessary to remove drug selection concurrent with establishing gametocyte cultures. Whilst gametocytes highly expressing the FRET sensor were observed and measurable, many gametocytes had low/no expression of the sensor.

### MitoTrackerRed CMXRos labelling of gametocytes
Gametocytes were transferred to a pre-warmed 1.5 ml tube in a 37 °C heater block and MitoTrackerRed CMXRos (hereafter MitoTracker)

**Fig. 8 | A model showing the potential contribution of mitochondrial ATP production to male gametocytes and male gametogenesis.** Under glucose-replete conditions, gametocytes can maintain intracellular ATP homeostasis even when mitochondrial ATP production is inhibited. However, male gametogenesis cannot complete without available glucose or mitochondrial ATP production.

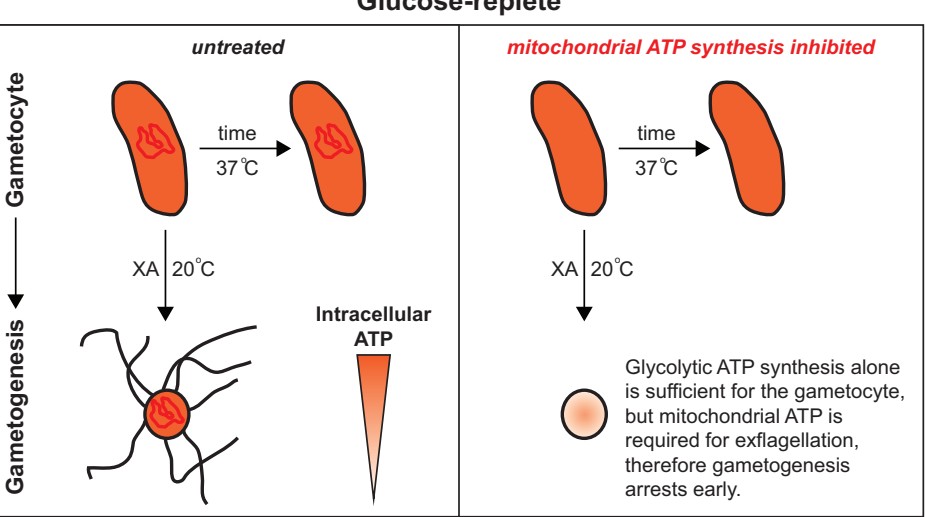

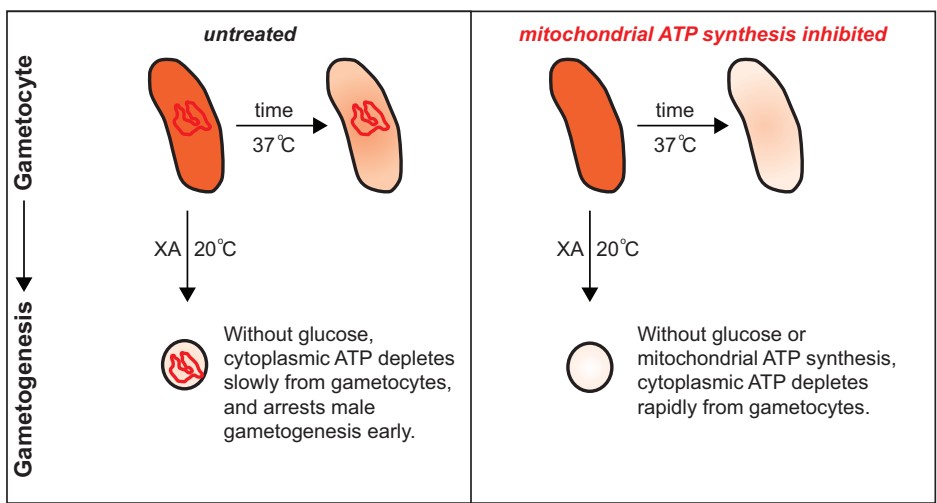

(ThermoFisher) added to the culture at the desired final concentration (200 nM for mitochondrial morphology, 12.5 nM for mitochondrial activity). Gametocytes were incubated with MitoTracker for 25 min then pelleted briefly in microcentrifuge and supernatant removed. Labelled gametocytes were washed twice with warm culture medium and then resuspended prior to fixing (see below).

### MitoTracker drug assay
Mature Day 14 gametocyte cultures were incubated at 37 °C for 24 h in gametocyte medium containing DMSO (drug vehicle control) or the following mETC inhibitors: (i) atovaquone (ATQ) (concentrations tested: 1 nM, 10 nM, 100 nM, 1 μM and 10 μM), (ii) ELQ-300 (concentrations tested: 1 nM, 10 nM, 100 nM, 1 μM and 10 μM), (iii) 10 μM DSM-265 and (iv) 50 μM oligomycin A. Gametocytes were subsequently labelled with 12.5 nM MitoTracker, fixed and then co-labelled with the anti-LDH2 antibody. Quantitative measurements of MitoTracker labelling were performed as described below.

### Immunofluorescence and fluorescence microscopy
Gametocytes for immunofluorescence assays (IFA) were fixed in culture medium 1:1 with 8% paraformaldehyde (PFA) in phosphate buffered saline (PBS) to give a final concentration of 4% PFA. Gametocytes were fixed for 30 min before 50 μl of the fixed cell solution was added to pre-prepared poly-lysine-coated coverslips in 500 μl PBS and cells were allowed to settle overnight at 4 °C. Thereafter, all IFA steps were carried out at room temperature. Fixed gametocytes were permeabilised with 0.1% Triton X-100 in PBS for 10 min and subsequently washed three times with PBS. Coverslips were blocked with PBS + 10% foetal bovine serum (FBS) (blocking buffer) for 30 min then stained for 1 h with primary antibodies diluted in blocking buffer. Rat anti-LDH2 was used at a 1:10,000 dilution; guinea pig anti-PfG377 at 1:10,000; mouse anti-Pfs230 (BEI Resources, MRA-878A) at 1:1000; anti-alpha tubulin (Merck, clone DM1A) at 1:500; anti-Pfs25 (BEI Resources, clone 4B7 – coupled to Cy3 fluorophore) at 1:500. Coverslips were then washed x3 with PBS before secondary antibody staining for 45 min in blocking buffer. Secondary antibodies used were raised against the respective species of the primary antibody coupled to a range of Alexa fluorophores (ThermoFisher), all diluted to 1:1000. After a final three washes with PBS, coverslips were then mounted on glass slides with DAPI-containing VectaShield (Vector Laboratories). Z-stack images of gametocytes were taken using a Nikon TiE Eclipse widefield microscope. Deconvolution and maximum intensity projections were performed within the Nikon NIS Elements analysis software.

### Quantitative image analysis
Images of gametocytes were processed in ICY BioImage Analysis software v2.5.2.0 (ICY) (https://icy.bioimageanalysis.org/). Individual images of gametocytes were cropped using the Rectangle + Fast crop tools. For the morphological analysis, a region of interest (ROI) was drawn around the

perimeter of the gametocyte using the Polygon tool. ROIs around the mitochondrion and nucleus were calculated by thresholding using the HK-Means plugin. Morphological parameters and staining intensities for MitoTracker and anti-LDH2 were exported in a comma separated values (csv) format using the ROI statistics plugin for downstream analysis. Gametocyte sex was determined by the intensity of anti-LDH2 staining which consistently separated gametocytes into two populations.

To calculate gametocyte mitochondrial activity as reported by total MitoTracker intensity within the mitochondrion, the total fluorescence intensity for the entire gametocyte was calculated. Then, by sampling a respective region of each individual gametocyte cytoplasm, the background fluorescence for the entire gametocyte was calculated and subtracted.

### Dual gamete formation assay (Pf DGFA)

Dual gamete formation assays were performed to compare the activity of gametocyte targeting compounds on the functionality of male and female gametocytes as reported by their ability to transform into male and female gametes when induced[24]. 384 well plates were prepared with dose responses of atovaquone, ELQ-300, oligomycin A and methylene blue, using DMSO and 20 µM Gentian Violet as negative and positive controls respectively. Mature Day 14 gametocyte cultures were dispensed into each well and incubated at 37 °C for 24 h. Then, gametogenesis was triggered by the addition of xanthurenic acid and cooling the plate to 28 °C. Simultaneously, a Cy3-labelled anti-Pf25 antibody was added to each well for the detection of female gametes. 20 min after induction, male gametogenesis was recorded by automated light microscopy of exflagellation centres, before incubating the plate at room temperature for a further 24 h. Female gametogenesis was then quantified by automated fluorescence microscopy by detecting and counting Pfs25-positive cells. % inhibition of each well was then calculated with reference to the positive and negative controls and dose response curves fitted in GraphPad Prism 9 to calculate $IC_{50}$ values.

### Manual exflagellation/female activation assays

To investigate the effect of varying serum concentrations, the presence and absence of glucose and/or oligomycin A, 100 µl samples of mature Day 14 gametocyte cultures were washed in warm gametocyte medium (without Albumax) of the appropriate glucose/serum condition for the experiment three times. Different medium glucose concentrations were achieved by preparing incomplete gametocyte medium with RPMI lacking glucose and supplementing back the appropriate concentration of glucose for the experiment (0–2 g/L glucose). Gametogenesis was then stimulated condition-matched medium at room temperature containing xanthurenic acid. If gametogenesis was to be quantified, samples were transferred to 96 well plates for automated imaging and quantification 20 min after triggering gametogenesis. If gametogenesis was to be visualised, samples were fixed after 20 min and processed as described for IFAs.

### FRET measurements of gametocyte cytoplasmic ATP level

Mature Day 14 gametocytes were enriched to ~80% purity by density gradient purification with Gentodenz® (Gentaur) in a heated centrifuge. Purified gametocytes were maintained at 37 °C in gametocyte medium in a tube rack of a heater block. Gametocyte medium without human serum, albumax or glucose was prepared (no-glucose medium), and 2 g/L glucose added back to reconstitute the medium (glucose medium) and oligomycin A added if necessary to a final concentration of 5 µM. PBS with or without 2 g/L glucose and with or without oligomycin A were also prepared and warmed to 37 °C. Enriched gametocytes were briefly pelleted in a microcentrifuge and washed twice with either glucose or no-glucose medium (with or without oligomycin A), and then washed a further two times in condition-matched PBS for imaging (to minimise autofluorescence from phenol red in the RPMI medium). Washed cells were immediately transferred to a pre-warmed glass slide, a coverslip placed on top and sealed with petroleum jelly. Slide-mounted cells were then immediately transferred to a Nikon TiE2 widefield microscope with the stage surrounded with an environmental chamber heated to 37 °C. Cells were imaged using the x60 oil objective and a field of view selected to contain as many gametocytes brightly expressing the FRET sensor as possible. Time taken to prepare samples and select field of view for experimentation was set at 3 min to ensure consistency between replicates. After the 3 min preparation period, cells were imaged every minute for a further 20 min. The CFP moiety of the FRET sensor was excited by a 435 nm LED light source. CFP emission was collected using a DAPI emission filter (435–485 nm) and FRET (YFP) emission collected using a GFP emission filter (500–550 nm).

Captured images were processed in ICY to determine the FRET ratio. The background autofluorescence from each image was subtracted to leave only fluorescence from the sensor. Individual gametocytes were manually traced as ROIs and fluorescence values exported as csv files using the ROI Statistics plugin. The FRET ratio was calculated by dividing the sum total fluorescence of the YFP emission and the CFP emission for each individual gametocyte. To only capture gametocytes highly expressing the sensor, those gametocytes with a mean CFP fluorescence of less than 150 units were not recorded. The mean FRET ratio of each image in the timelapse was calculated, with each replicate consisting of three separately recorded timelapses for each condition tested.

### Statistics and reproducibility

All statistical tests were performed using GraphPad Prism version 10. Two-tailed unpaired Student's $t$-tests were performed to compare the difference in nuclear and mitochondrial morphology parameters between male and female gametocytes, and the differences in mitochondrial activity after drug treatment. Two-tailed ratio-paired $t$-tests were performed to analyse the difference between pairs of data where control values between replicates varied substantially due to the unavoidable inherent variability of the experimental design. Experiments measuring nuclear and mitochondrial morphology contained 3–4 independent replicates with 48–139 gametocytes measured per sample. Experiments measuring mitochondrial activity with MitoTracker were performed with 3–5 independent replicates measuring 35–294 gametocytes per replicate. Experiments quantifying exflagellation levels of gametocytes under different conditions contained 3–5 independent experiments. Experiments measuring FRET sensor parasites were performed with 6–9 independent experiments recording data from at least three different fields of view at x60 objective per replicate.

### Reporting summary

Further information on research design is available in the Nature Portfolio Reporting Summary linked to this article.

### Data availability

The source data for main figures can be found in Supplementary Data 1. All other data are available from the corresponding author on reasonable request.

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

## Acknowledgements

This work was supported by a Medical Research Council Career Development Award (MR/V010034/1) awarded to M.J.D.; P.C.S. was supported by an MRC-LID studentship. M.T.F. was supported by a Medicines for Malaria Venture award (RD-21-1003). The funders had no role in study design, data collection and analysis, decision to publish, or preparation of the manuscript.

## Author contributions

P.C.S. experimental design and execution, writing of manuscript; E.A. experimental design, parasite culture support; M.T.F. experimental design and execution, parasite culture support; E.S. experimental design; J.P. experimental design; M.J.D. conceived the study, experimental design, execution writing of manuscript.

## Competing interests

The authors declare no competing interests.
