## [Transparent Peer Review file · Communications Biology]

Mitochondrial ATP synthesis is essential for efficient gametogenesis in *Plasmodium falciparum*

Corresponding Author: Dr Michael Delves

Version 0:

Reviewer comments:

Reviewer #1

(Remarks to the Author)

This is a simple but informative study that contributes to our understanding of mitochondrial activity and ATP production during gametocytogenesis and gametogenesis. The authors identified a previously undescribed marker for late stage male gametocytes and male gametes and used this marker as a probe in novel and seemingly very powerful assays to distinguish male and female gametocytes within the same culture for quantitative and qualitative analyses of mitochondrial activity and inhibition. The study concludes with a thorough analysis of the contribution of glycolytic and mitochondrial ATP production during male gametogenesis.

The data are presented clearly and the authors are careful to never overstate their conclusions. In fact, in most places, the limitations of their data are carefully discussed.

Some minor comments:

1) The anti-LDH2 antibody clearly distinguishes between male and female gametocytes. However, the authors never show any data validating that this antibody (or the anti-PfG377 antisera produced for this study) specifically recognizes the intended antigen. Something as simple as a western blot showing that the antibody only recognizes a single band of the predicted size of LDH2 would suffice. It may also be possible to show specificity by competing away the antibody using the peptides used for immunization in an IFA.

2) The "partial inhibition plateau" observed in female gametocytes treated with atovaquone and ELQ-300 (Figure 4A & B and discussed starting line ~216 of the text) is unexpected and I'm not sure I follow the explanation offered in the text. Are the authors suggesting that the true percent inhibition of female gametogenesis is closer to 0%, and ~35% of of female gametes are just expressing Pfs25 at levels below the limit of detection at plateau concentrations? Mitochondrial inhibition affecting protein abundance is a reasonable hypothesis, but I don't understand why it would plateau with ~65% still producing detectable protein levels even at concentrations that completely eliminate MitoTracker CMXRos staining. It would be nice to see this followed up in the "manual" female activation assay, though I'm not sure that is necessary given the scope of this study.

3) There are a few instances in the introduction where aspects of gametocyte biology are stated without citation that should really be cited (eg: line 34: "activating factor xanthurenic acid", line 74: "the parasite mitochondrion is female-inherited").

4) In Figure 3A, the second column of images are labeled "DAPI", though it looks as though these should be labeled "CMXRos".

Reviewer #2

(Remarks to the Author)

The authors described the importance of mitochondrial ATP synthesis for efficient gametogenesis in *Plasmodium falciparum*

clearly, which were supported by basic experiments as well as ATP biosensors. The paper pointed out the quite impressive feature in gametogenesis and give impactful insight for this research field. Yet, a few points should be addressed before publication.

- 1) For those who are not familiar with this research field, it is kind of hard to capture the life cycle of Plasmodium in the infectious process of mosquito-human. In particular, in the liver and blood, asexual and sexual gametogenesis, etc. The reviewer suggests preparing the brief schematic image to explain this introduction.
- 2) The author investigated the morphology of mitochondria in gametes in male and female as well as its activities. Why MitoTracker™ CMXRos is possible to evaluate the mitochondria activities? It meant that the mitochondrial membrane potential was examined indirectly based on the accumulation of the dye. It should be stated more clearly. Also, are there any specific reasons not to use TMRM (which is more sensitive to mitochondrial membrane potential)?
- 3) Surely, oligomycin A is known to inhibit the ATP synthase directly reducing the ATP production. Then, the mitochondrial membrane potential increases due to the inhibition, which is consistent with the results through the paper. On the other hand, the respiration goes down while the glycolysis is accelerated sometime to compensate the depletion of the decrease of cytoplasmic ATP. It is better to argue the alternative possibility regarding the contribution of glycolysis more. The data seems lacking the direct observation for the glycolysis, as the other important ATP synthetic pathway (ECAR, whatever).
- 4) Though it is the same point with 3), after the inhibition of mitochondrial ATP, the cytoplasmic ATP level scarcely changed in the presence of glucose. It seems kind of curious because oxphos is likely to contribute cytoplasmic ATP more or less though the glycolysis is major player in this case. Why cytoplasmic ATP level rarely changed?

Reviewer #3

(Remarks to the Author)

The authors address the urgent need to understand *P. falciparum* resistance to anti-malarial drugs, which target upregulated mitochondrial respiration that occurs during gametogenesis. *P. falciparum* rely on glycolysis-driven ATP production during the asexual stage and expand their respiratory capacity by upregulating mitochondrial networks during gametogenesis in preparation for transmission from human to mosquito hosts. The authors suggest that drug resistance occurs because *P. falciparum* exists in the asexual stage in the human host and undergoes gametogenesis (i.e., becomes more dependent on mitochondrial respiration) only during transmission to mosquitoes. The authors also address the apparently unwarranted investment in male gamete mitochondrial biogenesis, as *P. falciparum* mitochondria are female-inherited. The authors provide evidence that male mitochondrial network upregulation and respiration-associated ATP production is required for gametogenesis, and consequently protects them from respiration-targeting anti-malarial drugs. Overall, the authors provide convincing evidence for their results. To more definitively determine that male gametogenesis cannot be achieved using glycolytically produced ATP, authors need to report total glucose concentration in exflagellation experiments. Two mM glucose was added to glucose-free media in the ATP quantification experiments, but it is unclear what the glucose concentration was in the exflagellation experiments for which 0.1% human serum and glucose-containing media were used. This is critical in demonstrating that lack of exflagellation was not due to decreased gamete viability. I recommend that the manuscript be accepted for publication after the concerns detailed below are adequately addressed.

Because of a lack of training in these areas, recommendations for the following experiments are not provided:

- Identification of sex-specific proteins and antibodies that label them
- Gamete formation
- Manual exflagellation/female activation
- ATP biosensor sensitivity.

Minor

- Figure 2 caption: Remove “* = $p < 0.05$.”
- Figure 3A is not referenced in the text.
- Figure 3A appears to be mislabeled – the middle panel appears to MitoTracker rather than DAPI labeling.
- 3C shows 6 data points for males, but caption indicated 5 experiments were performed.

Quantitative image analysis method: Clarify whether maximal or average intensity of MitoTracker was reported.

Major

- Quantitative imaging of male and female gametocytes
 - o The ROI in the male image in Figure 2A appears to extend beyond the nucleus, and this extension lack mitochondrial that are visibly outside the ROI. This is a critical point as the authors report sex differences in nuclei morphology and size (2C). Address why male ROI appears to extend beyond nucleus and lack mitochondria.
 - o 2C Nuclear interior measurement appears to be underpowered.
 - o Results section indicates ROI nuclear elongation was reported, but these data are not represented in Figure 2.
 - o Figure 3 C indicated increased MitoTracker staining in females; however, this result is not reflected in the representative image in 3A.
 - o 3C appears to report exactly 100% for all female samples from four experiments consisting of 47, 19, 49, 59, and 35 gametocytes. To be forthcoming with variability among female samples, authors should graph non-normalized data. Alternatively, raw values could be divided by the average of female samples for each experiment. For example, if the average for females in experiment 1 was 90, then raw data from all 47 gametocytes for both sexes should be divided by 90.
- The effect of mitochondrial inhibitors on male and female gametocytes
 - o Figure 4A,B,D should have significance starts to indicate statistical differences between conditions.
 - o Figure 4A-D: Female variability should be reported (see previous comment).
- Inhibition of mitochondrial respiration by oligomycin A stalls male gametogenesis
 - o To confirm that exflagellation cannot be promoted by glycolytically produced ATP that could be present in typically culture (i.e., with serum concentrations between 10 and 20%), authors should perform experiments with higher concentrations of

human serum and increasing concentrations of glucose in the presence of oligomycin.

o To more definitively determine that male gametogenesis cannot be achieved using glycolytically produced ATP, authors need to report total glucose concentration in exflagellation experiments. Two mM glucose was added to glucose-free media in the ATP quantification experiments demonstrating viability, but it is unclear what the glucose concentration was in the exflagellation experiments for which 0.1% human serum and glucose-containing media were used. This is critical in demonstrating that lack of exflagellation was not due to decreased gamete viability.

Version 1:

Reviewer comments:

Reviewer #1

(Remarks to the Author)

The authors have taken care to specifically address my comments from the previous submission of this manuscript: an added supplementary figure showing the specificity of the antibodies generated for the study, and a clearer explanation for the observed inhibition plateau in female gametocytes treated with atovaquone and ELQ300.

This is a brief yet thorough study that provides new tools (antigens/antibodies) and a novel assay framework for studying one of the harder stages of the parasite lifecycle.

Reviewer #2

(Remarks to the Author)

The author addressed questions properly except the direct evidence of glycolysis. Yet, we could understand that it was not possible due to the limitation of category pathogen 3.

Reviewer #3

(Remarks to the Author)

Remaining minor suggestions regarding authors' response to two Major comments on initial submission:

1. Quantitative imaging of male and female gametocytes

o Initial comment: 3C appears to report exactly 100% for all female samples from four experiments consisting of 47, 19, 49, 59, and 35 gametocytes. To be forthcoming with variability among female samples, authors should graph non-normalized data. Alternatively, raw values could be divided by the average of female samples for each experiment. For example, if the average for females in experiment 1 was 90, then raw data from all 47 gametocytes for both sexes should be divided by 90.

• New comment: I recommend the authors replace Figure 3A with supplementary Figure 4A, which is much more descriptive, and thus informative.

2. The effect of mitochondrial inhibitors on male and female gametocytes

o Initial comments: (1) Figure 4A,B,D should have significance stars to indicate statistical differences between conditions. (2) Female variability should be reported.

• New Comment: If authors normalized by dividing each raw value, including female control raw values, by the average of control female raw values, they will be able to show variability in the control female condition and run statistical analyses. Otherwise, they cannot report a "dose-dependent inhibition" of female mitochondrial activity (line 207). Alternatively, they can run statistical analysis on data in supplementary figure 6. In this case, I recommend the authors replace figure 4 AB with supplementary figure 6 left panels.

For both minor points, I leave the decision on whether the authors address these comments to the editor, since these results do not affect the major conclusions. I recommend that the manuscript be accepted for publication.

Response to Reviewers' comments COMMSBIO-24-2978-T

We thank the reviewers for their encouraging comments and have endeavoured to answer them all where possible and provide justification if not fully addressed. Please see below for our responses inline with the comments. A tracked changes version of the manuscript is included however all line references in the response refer to the "clean" version.

Reviewer #1 (Remarks to the Author):

This is a simple but informative study that contributes to our understanding of mitochondrial activity and ATP production during gametocytogenesis and gametogenesis. The authors identified a previously undescribed marker for late stage male gametocytes and male gametes and used this marker as a probe in novel and seemingly very powerful assays to distinguish male and female gametocytes within the same culture for quantitative and qualitative analyses of mitochondrial activity and inhibition. The study concludes with a thorough analysis of the contribution of glycolytic and mitochondrial ATP production during male gametogenesis.

The data are presented clearly and the authors are careful to never overstate their conclusions. In fact, in most places, the limitations of their data are carefully discussed.

Some minor comments:

1) The anti-LDH2 antibody clearly distinguishes between male and female gametocytes. However, the authors never show any data validating that this antibody (or the anti-PfG377 antisera produced for this study) specifically recognizes the intended antigen. Something as simple as a western blot showing that the antibody only recognizes a single band of the predicted size of LDH2 would suffice. It may also be possible to show specificity by competing away the antibody using the peptides used for immunization in an IFA.

Please find now included a peptide-blocking IFA demonstrating that 1 mg/ml of the peptides used to immunise against LDH2 and PfG377 can completely block staining with the antibodies (**Supplementary Figure 2**).

2) The "partial inhibition plateau" observed in female gametocytes treated with atovaquone and ELQ-300 (Figure 4A & B and discussed starting line ~216 of the text) is unexpected and I'm not sure I follow the explanation offered in the text. Are the authors suggesting that the true percent inhibition of female gametogenesis is closer to 0%, and ~35% of of female gametes are just expressing Pfs25 at levels below the limit of detection at plateau concentrations? Mitochondrial inhibition affecting protein abundance is a reasonable hypothesis, but I don't understand why it would plateau with ~65% still producing detectable protein levels even at concentrations that completely eliminate MitoTracker CMXRos staining. It would be nice to see this followed up in the "manual" female activation assay, though I'm not sure that is necessary given the scope of this study.

We thank the reviewer for raising this important point and feel that perhaps our explanation of this phenomenon was not described clearly enough. The PfDGFA female readout for ATQ and ELQ300 clearly shows a classical dose response curve, however the curve plateaus out ~65% inhibition. Our hypothesis is that this phenomenon is a combination of both the cell biology of the female gamete

and the readout in our assay. Unlike the male gametocyte that undergoes rapid development during gametogenesis (i.e. requires a lot of energy in a short space of time), the female gamete (we think) needs only to egress from the erythrocyte and start expressing her translationally repressed mRNA - Pfs25 being one of these (i.e. energy provision is less time-sensitive). Therefore, like with the male, we envisage a situation where mitochondrial inhibition does not prevent the start of female gametogenesis, just limits its progression. We expect this to manifest in the female as a reduced rate of protein expression of Pfs25. We hypothesise that perhaps the plateau we observe reflects the drug concentrations where the gametocyte mitochondria is non-productive and what Pfs25 expression that is observed is fuelled by glycolysis alone. Under these circumstances, most female gametes fall below our assay-defined Pfs25 expression threshold, however some (~35%) still manage to express enough to be detected.

The discussion of these data has been clarified on **lines 220-229**.

3) There are a few instances in the introduction where aspects of gametocyte biology are stated without citation that should really be cited (eg: line 34: "activating factor xanthurenic acid", line 74: "the parasite mitochondrion is female-inherited").

The requested citations are now added to the manuscript – Billker et al 1998 (3) for identification of xanthurenic acid and Creasy et al 1994 (14) for maternal inheritance of the mitochondrion.

4) In Figure 3A, the second column of images are labeled "DAPI", though it looks as though these should be labeled "CMXRos".

We thank the reviewer for spotting this error. It is now corrected.

Reviewer #2 (Remarks to the Author):

The authors described the importance of mitochondrial ATP synthesis for efficient gametogenesis in *Plasmodium falciparum* clearly, which were supported by basic experiments as well as ATP biosensors. The paper pointed out the quite impressive feature in gametogenesis and give impactful insight for this research field. Yet, a few points should be addressed before publication.

1) For those who are not familiar with this research field, it is kind of hard to capture the life cycle of *Plasmodium* in the infectious process of mosquito-human. In particular, in the liver and blood, asexual and sexual gametogenesis, etc. The reviewer suggests preparing the brief schematic image to explain this introduction.

There are very many published diagrams of the parasite life cycle. We have added a reference on **line 30** (Venugopal et al 2020 (2)) to a review article that displays excellent summary diagrams of the *Plasmodium* life cycle.

2) The author investigated the morphology of mitochondria in gametes in male and female as well as its activities. Why MitoTracker™ CMXRos is possible to evaluate the mitochondria activities? It meant that the mitochondrial membrane potential was examined indirectly based on the accumulation of the dye. It should be stated more clearly. Also, are there any specific reasons not to use TMRM (which is more sensitive to mitochondrial membrane potential)?

We explain very clearly in the manuscript the rationale and activity of using MitoTracker™ Red CMXRos the first time it is mentioned:

Line 121: “Gametocytes were prelabelled with the dye MitoTracker™ Red CMXRos (hereafter, MitoTracker) which accumulates in functional mitochondria possessing a negative membrane potential and is retained after aldehyde fixation”.

TMRM is potentially another mitochondrial label that we could have used, however it is not suited to our specific use and we chose not to use it as it is not retained after aldehyde fixation. The imaging data presented in our manuscript are the result of 50+ hours of imaging. Imaging fixed cells labelled with MitoTracker™ Red CMXRos enabled us to capture a “snapshot” of activity. If the same experiments were conducted on live cells, firstly we would not have been able to use our male gametocyte-specific antibody to determine sex, but also there would be significant drift in data due to length of time taken to image so many live cells per experiment.

3) Surely, oligomycin A is known to inhibit the ATP synthase directly reducing the ATP production. Then, the mitochondrial membrane potential increases due to the inhibition, which is consistent with the results through the paper.

Correct, this is the hypothesis we present in the paper which accounts for the observed increase in MitoTracker™ Red CMXRos labelling shown in **Figure 4D**.

On the other hand, the respiration goes down while the glycolysis is accelerated sometime to compensate the depletion of the decrease of cytoplasmic ATP. It is better to argue the alternative possibility regarding the contribution of glycolysis more.

If I understand correctly, the reviewer is suggesting that in oligomycin A-treated cells, inhibiting mitochondrial respiration potentially causes glycolysis to compensate and try to raise cytoplasmic ATP levels? Whilst not tested directly in our manuscript, this is not in conflict with our observations. **Figure 7** shows that oligomycin A treatment in the presence of glucose does not alter cytoplasmic ATP levels. Our conclusion from this is that glycolysis alone is sufficient to maintain cytoplasmic homeostasis, whether it is raised in compensation or not.

The data seems lacking the direct observation for the glycolysis, as the other important ATP synthetic pathway (ECAR, whatever).

Agreed, direct observations of glycolysis supplementing our ATP sensor data would be highly valuable and this is definitely part of our intended future work. However measurements such as ECAR are prohibitively challenging when working with *Plasmodium* gametocytes due to the biomass and purity required for accurate measurement and the expense of specialised equipment such as the Seahorse XFp Analyzer which are usually a shared resource not generally accessible for those working with Category 3 pathogens. If further research funding is forthcoming, these experiments will be a high priority for us, but at the moment, they are unfortunately not possible for us to perform for this manuscript.

4) Though it is the same point with 3), after the inhibition of mitochondrial ATP, the cytoplasmic ATP level scarcely changed in the presence of glucose. It seems kind of curious because oxphos is likely to contribute cytoplasmic ATP more or less though the glycolysis is major player in this case. Why cytoplasmic ATP level rarely changed?

As we discuss on **lines 378-396 and 422-435 and Figure 8**, our hypothesis for the lack of change in

cytoplasmic ATP level after mitochondrial inhibition is that gametocytes, (being a relatively dormant and inactive stage of the parasite's life cycle) simply do not need mitochondrial ATP production to meet their needs – the main reason they keep a mitochondrion is because it is absolutely essential for mosquito transmission and the parasite cannot complete its life cycle without it.

Reviewer #3 (Remarks to the Author):

The authors address the urgent need to understand *P. falciparum* resistance to anti-malarial drugs, which target upregulated mitochondrial respiration that occurs during gametogenesis. *P. falciparum* rely on glycolysis-driven ATP production during the asexual stage and expand their respiratory capacity by upregulating mitochondrial networks during gametogenesis in preparation for transmission from human to mosquito hosts. The authors suggest that drug resistance occurs because *P. falciparum* exists in the asexual stage in the human host and undergoes gametogenesis (i.e., becomes more dependent on mitochondrial respiration) only during transmission to mosquitoes. The authors also address the apparently unwarranted investment in male gamete mitochondrial biogenesis, as *P. falciparum* mitochondria are female-inherited. The authors provide evidence that male mitochondrial network upregulation and respiration-associated ATP production is required for gametogenesis, and consequently protects them from respiration-targeting anti-malarial drugs. Overall, the authors provide convincing evidence for their results.

To more definitively determine that male gametogenesis cannot be achieved using glycolytically produced ATP, authors need to report total glucose concentration in exflagellation experiments.

Apologies for this omission. We have now explicitly included the total glucose concentration of our standard “gametocyte medium” (4g/litre) in the materials and methods (**lines 491-503**). This gametocyte medium recipe was used routinely throughout the manuscript, however for the experiments specifically studying glucose and serum concentrations, we used varying concentrations of glucose depending on what was being tested. The materials and methods for each experiment state the concentration of glucose used, however we have now restated this in case it was not clear.

Two mM glucose was added to glucose-free media in the ATP quantification experiments, but it is unclear what the glucose concentration was in the exflagellation experiments for which 0.1% human serum and glucose-containing media were used. This is critical in demonstrating that lack of exflagellation was not due to decreased gamete viability.

Apologies if this was not clear - 2 g/L glucose was used in the exflagellation experiments with the serum titration. The legend for figure 6 has been updated to reflect this and also on **lines 306-315** in the text.

I recommend that the manuscript be accepted for publication after the concerns detailed below are adequately addressed.

Because of a lack of training in these areas, recommendations for the following experiments are not provided:

- Identification of sex-specific proteins and antibodies that label them
- Gamete formation
- Manual exflagellation/female activation
- ATP biosensor sensitivity.

Minor

- Figure 2 caption: Remove “* = p<0.05.”

Removed

- Figure 3A is not referenced in the text.

Corrected.

- Figure 3A appears to be mislabeled – the middle panel appears to MitoTracker rather than DAPI labeling.

Corrected.

- 3C shows 6 data points for males, but caption indicated 5 experiments were performed.

Corrected.

Quantitative image analysis method: Clarify whether maximal or average intensity of MitoTracker was reported.

As detailed on lines 556-567 it was the total (i.e. sum intensity) of MitoTracker staining that was calculated: “mitochondrial activity” was calculated by first recording the total MitoTracker staining within a deconvolved maximum intensity projection image of a gametocyte. The mean MitoTracker fluorescence of a representative area of the same gametocyte that did not contain the mitochondrion was then multiplied by the total area of the gametocyte to estimate the “background” fluorescence of the gametocyte. This value was then subtracted from the total MitoTracker staining to calculate the final “mitochondrial activity” value.

Major

- Quantitative imaging of male and female gametocytes
 - o The ROI in the male image in Figure 2A appears to extend beyond the nucleus, and this extension lack mitochondrial that are visibly outside the ROI. This is a critical point as the authors report sex differences in nuclei morphology and size (2C). Address why male ROI appears to extend beyond nucleus and lack mitochondria.

As described in the materials and methods (line 560), both the nucleus and mitochondria ROIs were automatically generated using an automated “HK-means” thresholding algorithm (<https://icy.bioimageanalysis.org/plugin/hk-means/>) which uses K-means clustering on the image channel histogram to divide the image into areas within or outside the ROI. Images are captured and analysed at 16-bit, but need to be resampled down to 8-bit for reproduction in the manuscript. This leads to low fluorescence staining (but still specific staining) being not readily visible in the 256 shades of grey possible in 8 bit images (compared to 65,536 shades in 16-bit). To illustrate the specificity of the ROIs, they are now reproduced at enhanced histogram LUT settings as Supplementary Figure 3 in the supplementary data.

- o 2C Nuclear interior measurement appears to be underpowered.

Agreed, and likely this contributes to the lack of statistical significance.

- o Results section indicates ROI nuclear elongation was reported, but these data are not represented in Figure 2.

Apologies, this graph was accidentally omitted. It has been added back to Figure 2.

o Figure 3 C indicated increased MitoTracker staining in females; however, this result is not reflected in the representative image in 3A.

As discussed, this is an issue with reproducing 16 bit images as 8 bit images for publication. Using the exact same methodology detailed in our manuscript, the female gametocyte mitochondrion displayed in **Fig 3** is calculated to have “mitochondrial staining” of 4,346,931 arbitrary units. The male gametocyte mitochondrion has 3,825,303 – less than the female gametocyte and well within the experimental variation shown in **Supplementary Fig 4A+B**.

o 3C appears to report exactly 100% for all female samples from four experiments consisting of 47, 19, 49, 59, and 35 gametocytes. To be forthcoming with variability among female samples, authors should graph non-normalized data. Alternatively, raw values could be divided by the average of female samples for each experiment. For example, if the average for females in experiment 1 was 90, then raw data from all 47 gametocytes for both sexes should be divided by 90.

As explained on **line 178**, “Absolute levels of mitochondrial staining intensity varied between independent experimental replicates, however male gametocytes on average possessed 44.4% less mitochondrial activity than females (Ratio-paired t-test; $p = 0.002$) (Fig. 3C)”. Due to inherent variability between experiments with MitoTracker labelling/staining conditions, we conducted the Ratio Paired T-test to analyse the differences between males of females between each replicate to determine statistical significance. To present the bar graph in Figure 3C, we then normalised the data to the female.

As requested by the reviewer, the raw values for each replicate have been divided by the mean female value for each replicate to display the variability. This is now presented as **Supplementary Fig 4A alongside the existing montage of individual images of Mitotracker-labelled gametocytes also illustrating the variability between samples**.

- The effect of mitochondrial inhibitors on male and female gametocytes
- o Figure 4A,B,D should have significance starts to indicate statistical differences between conditions.
- o Figure 4A-D: Female variability should be reported (see previous comment).

As recommended by the reviewer, we have normalised all raw data to the mean of the DMSO control female values for each experimental replicate in the left column of **Fig. 4**. Given that this means that all female DMSO controls are 100, only a statistical test (Unpaired Student’s T-test) is performed to compare the male DMSO and male experimental value at the highest concentration tested. ATQ, ELQ-300 and oligomycin A all show a significant change. The variability of all experimental data from the mitochondrial experiment now can be visualised in **Supplementary Fig 6**.

- Inhibition of mitochondrial respiration by oligomycin A stalls male gametogenesis
- o To confirm that exflagellation cannot be promoted by glycolytically produced ATP that could be present in typically culture (i.e., with serum concentrations between 10 and 20%) , authors should perform experiments with higher concentrations of human serum and increasing concentrations of glucose in the presence of oligomycin.

All of the exflagellation assays and mitochondrial activity assays presented in Figure 4 were performed in Gametocyte culture medium containing 5% human serum and 4 g/L glucose (twice the typical glucose concentration of RPMI and much higher than typical physiological glucose serum concentrations of ~70-100 mg/L). In addition, we now include as **Fig. 5B** and discuss in the text on **lines 290-295**, a dose response of the effect glucose concentration on exflagellation in medium containing 0.1% human serum. This gives an IC₅₀ glucose concentration of 19 mg/L which fully saturates by 52 mg/L – seventy seven times less than routinely used in our assays. Whilst it is theoretically possible that increasing glucose concentrations further could compensate for oligomycin A inhibition of mitochondrial ATP generation, it would require a highly non-physiological glucose concentration which may start to have a negative effect by affecting the osmotic properties of the culture medium.

o To more definitively determine that male gametogenesis cannot be achieved using glycolytically produced ATP, authors need to report total glucose concentration in exflagellation experiments. Two mM glucose was added to glucose-free media in the ATP quantification experiments demonstrating viability, but it is unclear what the glucose concentration was in the exflagellation experiments for which 0.1% human serum and glucose-containing media were used. This is critical in demonstrating that lack of exflagellation was not due to decreased gamete viability.

To clarify, 2mM glucose was not used in experiments, 2 g/L (approximately 11.1 mM) was used in these experiments. Please refer to previous comments. The glucose concentration used in specific experiments is now clarified in the results and materials and methods sections which hopefully avoids confusion.

Remaining minor suggestions regarding authors' response to two Major comments on initial submission:

1. Quantitative imaging of male and female gametocytes

o Initial comment: 3C appears to report exactly 100% for all female samples from four experiments consisting of 47, 19, 49, 59, and 35 gametocytes. To be forthcoming with variability among female samples, authors should graph non-normalized data. Alternatively, raw values could be divided by the average of female samples for each experiment. For example, if the average for females in experiment 1 was 90, then raw data from all 47 gametocytes for both sexes should be divided by 90.

• **New comment: I recommend the authors replace Figure 3A with supplementary Figure 4A, which is much more descriptive, and thus informative.**

As recommended by Reviewer 3, Figure 3A has now been replaced with Figure 4A (which has now been removed from the supplementary files).

2. The effect of mitochondrial inhibitors on male and female gametocytes

o Initial comments: (1) Figure 4A,B,D should have significance starts to indicate statistical differences between conditions. (2) Female variability should be reported.

• **New Comment: If authors normalized by dividing each raw value, including female control raw values, by the average of control female raw values, they will be able to show variability in the control female condition and runs statistical analyses. Otherwise, they cannot report a “dose-dependent inhibition” of female mitochondrial activity (line 207). Alternatively, they can run statistical analysis on data in supplementary figure 6. In this case, I recommend the authors replace figure 4 AB with supplementary figure 6 left panels.**

As recommended by Reviewer 3, the mitochondrial activity graphs of Figure 4 (previously A-D but now labelled A, C, E, G have been replaced with the data that was presented in Supplementary Figure 6 (this supplementary figure has now been removed from the supplementary files). Data presented is normalised to the DMSO control female mitochondrial activity for each drug treatment (allowing statistical comparison of the male gametocyte data). However, an alternative normalisation was also performed against the DMSO control of the male mitochondrial data, allowing statistical comparison of the female data as well (as explained in the figure legend). The raw data and both of the two normalised datasets are included in the Supplementary Data 1 file. Where there is a statistically significant difference from the DMSO controls, the p values are printed on Figure 4 above the respective data.